# A Multilevel Multiresolution Machine Learning Classification Approach: A Generalization Test on Chinese Heritage Architecture

**Kai Zhang** [1], **Simone Teruggi** [1,*], **Yao Ding** [2] and **Francesco Fassi** [1]

1   3D Survey Group, ABC Department, Politecnico di Milano, Via Ponzio 31, 20133 Milano, Italy
2   Institute of Architectural History and Theory, Tianjin University, Road Weijin 92, Tianjin 300072, China
*   Correspondence: simone.teruggi@polimi.it

**Abstract:** In recent years, the investigation and 3D documentation of architectural heritage has made an efficient digitalization process possible and allowed for artificial intelligence post-processing on point clouds. This article investigates the multilevel multiresolution methodology using machine learning classification algorithms on three point-cloud projects in China: Nanchan Ssu, Fokuang Ssu, and Kaiyuan Ssu. The performances obtained by extending the prediction to datasets other than those used to train the machine learning algorithm are compared against those obtained with a standard approach. Furthermore, the classification results obtained with an MLMR approach are compared against a standard single-pass classification. This work proves the reliability of the MLMR classification of heritage point clouds and its good generalizability across scenarios with similar geometrical characteristics. The pros and cons of the different approaches are highlighted.

**Keywords:** cultural heritage; point cloud; classification; machine learning; Chinese architecture

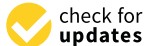



## 1. Introduction

### 1.1. Three-Dimensional Documentation and Artificial Intelligence in Cultural Heritage

Thanks to the rapid development of laser scanners and photogrammetric techniques, 3D modelling post-processing is conducted to make use of the digitized spatial data. This mostly consists of manual and time-consuming operations. Furthermore, these operations reduce the metric quality of the data, as the simplification is not always dictated by the purpose but rather by the limitations of the software and is subjective depending on the technical skills and personal interpretation of the operator. A potential solution able to minimize the time consumption and thus the costs of these modelling processes is using the point clouds directly during the production process [1]. Previous research has focused on developing strategies that allow these point models to be used as triangular meshes, enabling their use inside augmented-reality (AR) and mixed-reality (MR) applications in cultural heritage (CH) [2–4]. However, point models, being rich in metric content, lack semantic meaning in their parts. This is an essential feature for allowing their use as "real" 3D models. The semantic classification and segmentation of the 3D dataset are mandatory steps that assign semantic meaning to each point through their classification into predetermined classes.

Manual segmentation requires an expert operator to visually interpret the dataset, subdividing it into relevant elements and grouping together points belonging to the same element. This process is as difficult, time-consuming, and subjective as the modelling phase itself. Recent advancements in artificial intelligence (AI), machine learning (ML), and deep learning (DL) provide a solution for managing point cloud datasets in a more rational and semi-automatic way. Their potential has already been proved in many other fields, such as in natural language processing and image classification (computer vision), with examples such as AlexNet [5] and ResNet [6]. Algorithms have been developed to classify the datasets distributing semantic meaning to each segment. Indeed, the labour

involved in defining each item one by one can be largely reduced. Actual applications of different techniques on the point cloud have varying performances and have proved essential to completing the classification task. Depending on the scenes analysed, neither of the methods (DL or ML) could generally outperform the other [7,8].

While the production of the 3D point cloud has become increasingly convenient, with precise and reliable results, AI is expected to play an important role in dealing with trillions of points (samples) collected from a complicated real scene.

### 1.2. Aim and Content

In this study, we apply and evaluate a classification method based on multilevel multiresolution (MLMR) combined with ML algorithms, testing its competence on singular databases and its generalizability from one to another.

Three ancient wooden structures were selected to test the validity and generalization of the approach: the great hall of Nanchan Ssu (China), the East Great Hall of Fokuang Ssu (China), and the Bell Tower of Kaiyuan Ssu (China). Different scales, similar architectural styles, and different building types can be helpful when discussing relevant classification topics: approaches, features, and, in particular, the generalization test, which is more efficient when it is controllable. In the section "Case studies", a brief description of the three architectures is given. In the Methodology, the MLMR classification approach and its results on the three cases studies are illustrated. In the Discussion, the behaviour of the MLMR approach compared with the non-hierarchical approach, the generalization test results, and the effect of the selection and computation of geometrical features are discussed. This work attempts to empirically evaluate the general pre-trained model for the MLMR classification of unseen datasets.

## 2. State of the Art

Based on segmentation and classification techniques, point clouds can be successfully exploited and better comprehended [9]. To classify and segment a point cloud model refers to the action of grouping points in subsets (commonly called segments) characterized by sharing one or more characteristics (geometric, radiometric, etc.). Segmentation methods could be grouped into edge-based, region growing, model fitting, hybrid, and AI (ML and DL). The last either relies on a set of provided training examples with manually annotated labels to learn how to perform the classification tasks (supervised learning) or seeks to build models that automatically understand how the data are organized (unsupervised learning). These approaches are generally robust against noise and occlusions but require a large amount of training data and high computing power to run the algorithm. As a result of the classification process, points were predicted for specific architectural elements, each being assigned a specific label that belongs to a set of previously defined classes.

In a supervised ML approach, including support vector machines (SVM) [10], random forest (RF) [11,12], and naïve Bayes [13], semantic categories are learned from a subset of manually annotated data that are used to train the classification model. This trained model is then used to spread the semantic classification to the entire dataset. Normally, it is not necessary to provide a large amount of annotated data for the training process to be effective.

More traditional methods, instead, typically rely on a range of hand-crafted shape descriptors as feature vectors from which to learn the classification pattern. These descriptors include local surface patches, spin images, intrinsic shape signatures, and heat kernel signatures [14]. A multiscale and hierarchical feature extraction method was introduced to obtain robust and discriminative characteristics [15]. Grilli et al. tested the extraction and the importance of geometric covariance features within the classification process; tests proved their validity in different case studies [16].

Unsupervised approaches differ from supervised learning approaches in that features themselves are learned as part of the training process. In recent years, the use of big data has made these methodologies, especially DL, accessible and popular. Among many

other methods, convolutional neural networks (CNNs) constitute the most representative approach in DL. Developed for 2D image analysis, CNNs have proved effective in different fields, e.g., object detection and model matching in street-view scenarios [17]. The progress made with 2D images acts as a foundation upon which to develop many 3D learning algorithms.

Among the different DL approaches, the main models for feature learning with the raw point cloud as input can be generally divided into point-based and tree-based approaches [18]. The first directly takes the raw point cloud as the input for training the DL network. The second employs a k-dimensional tree (Kd-tree) structure to transform the point cloud into a regular representation (linear representation afforded by the group action) and then feeds this into DL models.

*Classification in CH*

When dealing with a specific point cloud model in CH, it is necessary to meet the specific needs of that particular heritage. The performance of classification techniques for singular unique objects varies with their mass, material, color, surface variation, etc., and these features are used to train the classification model.

A 2.5D approach utilizes features and labels from 2D images, projecting them onto 3D models to perform classification. Texture-based classification [19,20] works with 2D data. The classification is performed on the texture image and on orthoimages obtained from the model. The results are then reprojected on the 3D model. For each case under study, optimized models, orthoimages, and UV maps are created. Semantic photogrammetry [21] uses the DL results of 2D images for 3D reconstruction to obtain a labelled 3D model.

HERitAge by point Cloud procESsing for Matlab (M_HERACLES) [22] performs segmentation from the scale of a historical neighbourhood up to that of the architectural element. The toolbox performs segmentation from the scale level of a neighbourhood to that of individual buildings using GIS shapefile data to assist the process. Afterwards, segmentation is performed from a building's scale level to that of architectural elements such as pillars and beams, utilizing several Euclidean geometry-based rules and slicing to identify clusters.

AI approaches that directly work on the 3D CH point model have only started to appear in recent years. An example of an ML approach that can work directly on the point cloud model is presented in [23]. This approach is based on the use of the RF algorithm and works on a set of manually labelled samples, with computed geometric covariance features. The model to be trained is fed with a manually defined training set, and it generates predictions on the previously segmented evaluation set to calculate its performance. Afterwards, the classification is spread to the whole dataset.

Starting from this point, a MLMR approach [24] works hierarchically on specific portions of the whole dataset, classifying it at different resolutions with increasing detail as the number of classes increases; this method has proved to be computationally economic and has allowed higher accuracy to be achieved on more complex architectural heritage buildings. Initially, the dataset is subsampled to a lower resolution (depending on the dimensions of the considered case study). Training a specific RF classifier, big macro-elements are classified. The result is then back interpolated on a point cloud of higher resolution so as to subdivide the elements that require higher geometric accuracy. The process iterates up to the classification of the full-resolution dataset (initial resolution). In contrast to the non-hierarchical approach, specific RF models are trained for each classification, but only a small number of labelled samples are required. The data are hierarchically split into sub-classes, while the level of geometric detail increases, allowing the discernment of architecture components processed on a limited portion of the dataset at a relevant resolution. The validity of the approach has been previously proved on Chinese wooden architecture [25]. The application of a single machine learning model across large and variable architectural datasets has been extensively tested, but the generalization ability

in the hierarchical approach requires further discussion, continuing from that discussed in [8,26].

In contrast to ML approaches, and based on the prospering of point cloud datasets, recent years have seen the application of DL networks to point model classification; examples include PointNet [27], PointNet++ [28,29], PCNN [30], and DGCNN [31]. DL frameworks based on these neural networks have been applied to the digital cultural heritage domain. Consequent applications, such as the improved DGCNN [32,33], support features such as normal and HSV colours coupled with the x, y, and z coordinates of the points. These approaches have proved to be effective, and the ArCH (Architectural Cultural Heritage) dataset [34] has been used to gather different pre-classified examples to train the network. A limitation is that the number of classes must be predetermined and constant across all models in the training dataset. However, due to the uniqueness and vastness of different heritage building examples, it is currently still very difficult, if not impossible, to define a dataset that has an adequate number of pre-classified examples to cover a complete range of heritage buildings. However, some studies have used transfer learning [35,36] against this deficiency. Cao et al. developed a DL approach that, using a pretrained example, is able to reduce the need for a bigger pre-segmented dataset, obtaining encouraging results [37–39].

## 3. Case Studies

Nanchan Ssu (NCS) and Fokuang Ssu (FKS), together with other temples and monasteries in Wutai Monti in the middle of China, constitute a UNESCO site, which was inserted into the World Heritage List in 2009. They date back to the late VIII, mid-VIII century. The bell tower of the Kaiyuan Ssu (KYS) could date back to the XI century. All three cases share the architectural style of the Tang dynasty.

All of the cases were surveyed with TLS (Trimble TX8) and UAV (DJI Phantom 4 Pro, DJI FC6310) by the members of the Archaeology Centre for Architecture, Settlement and Landscape (ACASL), Tianjin University, China.

The NCS temple was surveyed in 2017. The dataset of the great hall consists of a point cloud project produced from 78 scan stations (13 of which were inside the building). The entire dataset contains 1026.4 million points. The portion of the great hall (at 5 mm resolution) selected from the surroundings was subsampled to an average and uniform resolution of 15 mm (Table 1).

**Table 1.** Three-dimensional heritage point cloud data.

| Scene | Classes | Points * (mil) | Points Used (mil) | Mean Res. (mm) | Subs. Res. (mm) |
|---|---|---|---|---|---|
| NCS | 18 | 1026.4 | 354,229,166 | 5 | 15–60 |
| FKS | 18 | 2670.8 | 24,238,232 | 30 | 30–120 |
| KYS | 19 | 516.9 | 156,697,160 | 5 | 15–60 |

* Points number is referring to the whole monastery area.

The FKS temple was surveyed several times from 2015 to 2020. The dataset of the east great hall is obtained from a point cloud project of 2670.8 million points, resulting from 179 scans, among which 51 are interior scans. Due to the fact that the data acquisition method, the terrestrial laser scan (TLS), is highly limited within this project, especially within the roof structure, the used dataset is subsampled into one with a mean resolution of 30 mm, containing 24.2 million points (Table 1).

The KYS temple was surveyed from 2006. The dataset consists of a point cloud project produced from 39 scan stations (11 inside the building). The entire dataset contains 516.9 million points, and the portion of the bell tower is of 5 mm resolution (Table 1).

### 3.1. The Great Hall of Nanchan Ssu

Nanchan Ssu (NCS) (Figure 1) is located in Lijiazhuang, south-east of Wutai in Shanxi Province. The great hall, built in the Tang dynasty on a 1.2 m-high platform within the complex, is known as the earliest existing wooden structure in China. The earliest recording

written under the west beam of the central bay proves that the hall was reconstructed in 782 AD.

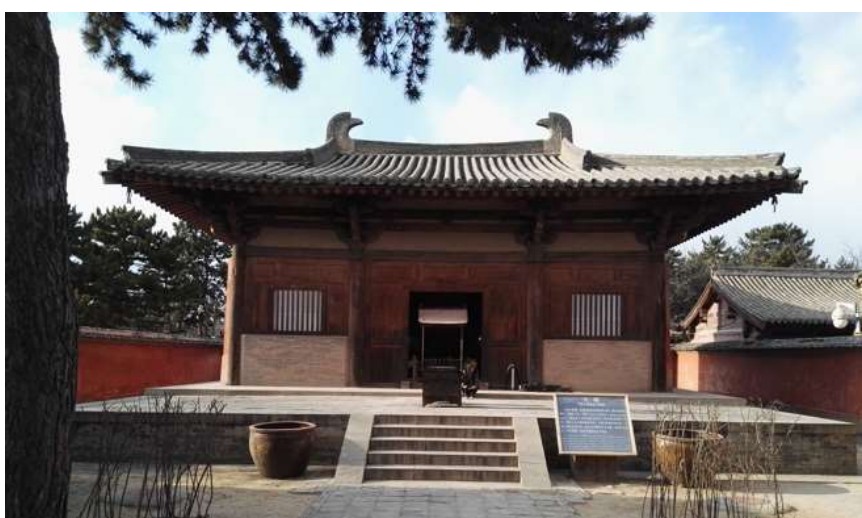

**Figure 1.** The front façade of the great hall of NCS (photo provided by ACASL, China).

It is a one-floor building, 9.2 m high, with a rectangular (10.5 × 11.7 m) floor plan divided into nine bays marked by pillars and beams. An altar of 0.7 m was built in the central bay, above which originally stood 17 painted statues (of which 14 statues remains today). The roof is supported by 10 wooden pillars, sealed with brick walls, in which two wooden windows and a door have been opened. During the intervention in the 1980s, two additional poles were placed inside to support the beams.

### 3.2. The East Great Hall of Fokuang Ssu

Fokuang Ssu (FKS) (Figure 2) is located northeast of Wutai County, Shanxi province, 6 kilometres from the town of Doucun, under the mountain of Fokuang (the west foot of the south part of the Wutai mount). From the inscription on the stone column in front of the great hall, the founding of the east great hall dates back to 857 AD.

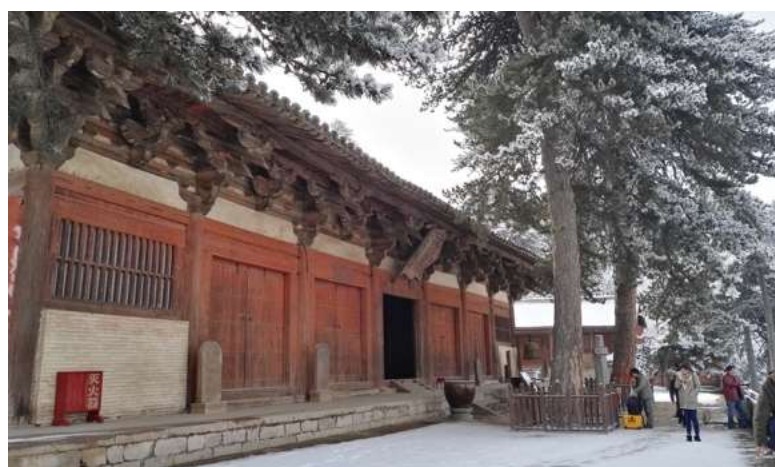

**Figure 2.** The front façade of the east great hall of FKS (photo provided by ACASL, China).

The east great hall was built on a 1.2 m high platform. It is a single-level construction, 15.4 m high, with a rectangular (34 × 17.66 m) floor plan. The 0.9 m high altar was carved from the stone mountain body under the five central bays, above which stand 35 painted tall statues that reach from 1.95 to 5.3 m in height. In the surrounding bays, it is possible to find other 296 statues. The roof is supported by 32 wood pillars, sealed with brick walls and with four wooden windows and five doors.

In a recent intervention, two additional poles were added to support the rear corners of the roof. The elements of the 'Dougong' (bracket sets) that hold up the roof have a cross section measuring 0.21 × 0.3 m, stretching the roof out from the main body of the hall of approximately 4 m.

This structure is one of the earliest architectures above ground that is still standing, being of great value to the Chinese nation and the East Asian cultural zone of architecture.

### 3.3. The Bell Tower of Kaiyuan Ssu

Kaiyuan Ssu (KYS) (Figure 3) is located in the southwest of the historical centre of Zhengding county, Hebei province. The temple could date back to 540AD; the bell tower shows strong features of the Tang dynasty, although there is no evidence to provide an accurate build time.

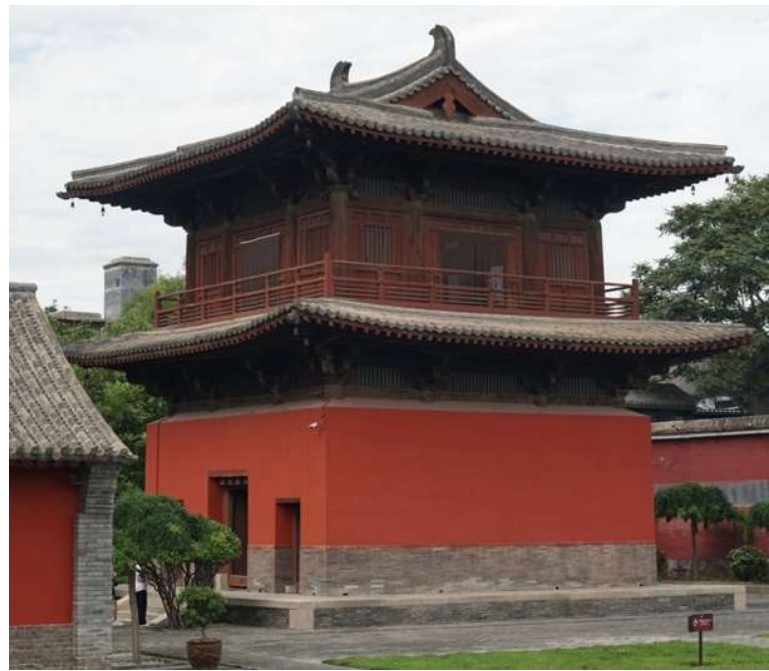

**Figure 3.** The bell tower of the KYS (photo provided by ACASL, China).

The east great hall was built on a 0.69 m high platform. It is a two-storey construction 14.5 m high, with a rectangular (9.8 × 9.8 m) floor plan divided into nine bays marked by four rows of pillars. Two stories are supported by 12 wooden pillars each. The ground level is sealed with brick walls with three wooden doors, while the upper level is mounted, with four doors and eight windows. Characteristically, between and under the bracket sets lay small windows on both levels. On the upper level stands a bronze bell of the Tang dynasty, hanging delicately on a wood frame, which is partly placed upon the main structure of the tower. In a recent intervention, the upper level was reconstructed following the Tang style.

### 4. Methodology

The projects under documentation have very large dimensions. They are rich in detail and have a high variety of architectural elements. The complexity and uniqueness of the three cases cause the unsupervised classification approach, especially the DL one, to not be applicable. It would be difficult and time-consuming to prepare enough representative samples upon which to train the model. Additionally, the conditions are the same in the non-hierarchical ML classification approach; the processing of such huge datasets requires many computational resources, both to extract the necessary geometric features for training and to make predictions on the whole dataset. Furthermore, a non-hierarchical classification with a large number of semantic classes would easily result in low accuracy.

In this work, an MLMR approach, as presented by Teruggi et al. [24], is applied in the study of three Chinese CH architectures (NCS, FKS, and KYS). The behaviours were compared with those of the non-hierarchical approach. In order to further test the generalization of the classifier, the RF model was trained by manually extracting annotated samples from the datasets of NCS and FKS. The model was later used to apply the classification to KYS.

In the MLMR approach, firstly, a set of multilayered semantic classes organized into a tree-like cornice is designed. Each class that defines macro-architectonic elements is subdivided into classes that can better describe every portion of these objects. In this way, the classification work is distributed across different levels, where a corresponding and relatively lower resolution diminishes the burden on the extraction of geometric features and training. The classification results are interpolated back to the same portion of the point cloud at higher resolution, which depends on the amount of geometric detail needed to discern the architectonic elements of this level. The segments allow the iterative classification and interpolation of the whole dataset, until specific portions reach full resolution. Compared with other common classification solutions, the MLMR approach is more accurate and computationally efficient for large-scale datasets.

For each MLMR classification, the datasets were first pre-processed ("noise" manually removed, elements not pertaining to the building cleaned, and the original point clouds subsampled), then geometric features were computed and appended to the datasets. Afterwards, and the training sets and evaluation sets were manually extracted and annotated with labels following the designed categories in order to train and evaluate the model. If performances were acceptable, then the model was used to predict classification on the whole dataset.

In the generalization tests, the same features at the same search radii were computed for two datasets, A and B. The test uses training sets and evaluation sets from dataset A to perform prediction on dataset B. The prediction results were compared with the ground truth, evaluating the performance of the model (Figure 4).

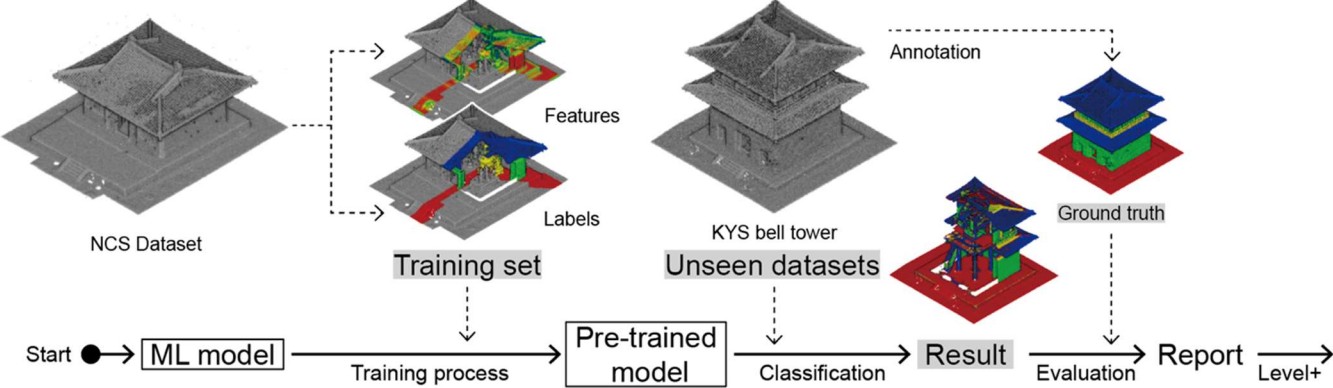

**Figure 4.** Classification scheme regarding the generalization process.

Covariance geometric features are required by the ML RF classifier in order to distinguish different elements. These features were computed using CloudCompare's [40] functions. Anisotropy, planarity, linearity, surface variation, sphericity, verticality, and normal vectors in the X and Y directions were the parameters used. It has been demonstrated that these are the geometric covariance features that most strongly affect the classification process [24]. These parameters were computed at different radii, which were chosen based on the measures of under-defining elements, depending on the case study being classified, as highlighted in Figure 5.

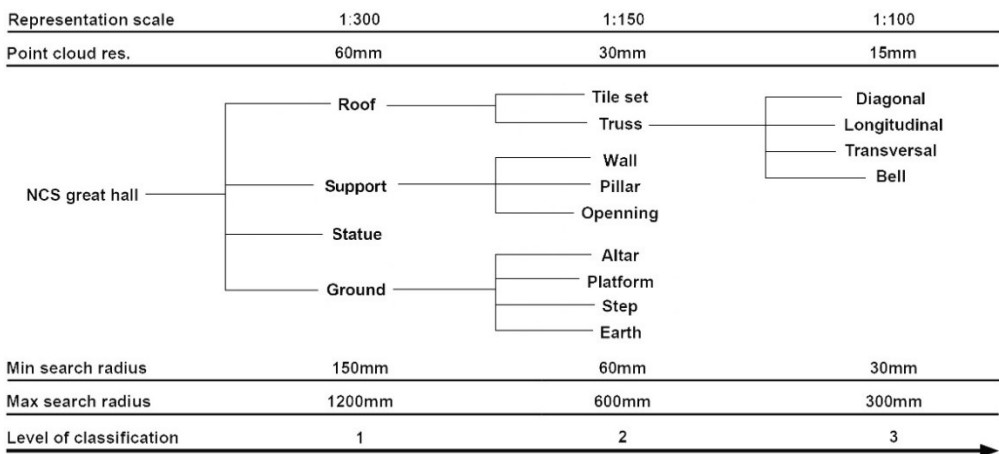

**Figure 5.** Point cloud classes for the NCS project.

### 4.1. The Great Hall of Nanchan Ssu

The classification of Nanchan Ssu's great hall is divided into three different levels, considering the monitoring requirements (from the overall composition to architectonic elements) as well as the original point cloud resolution (Figure 5).

The first level refers to the basic framework of the hall—roof, support, statue, and ground—vertically dividing the whole dataset into four classes. To recognize these categories, the classification was performed on a 60 mm resolution version of the entire dataset. The second level deepens these categories. The roof part was divided into tile set and truss. The support category comprises pillars, windows, and wall subsets. The ground was divided into altar, platform, steps, and earth. At this level, all elements were processed from a 30 mm resolution subsampled point cloud. The third level of classification was performed to classify the truss into longitudinal, transversal, and diagonal elements, using the dataset at its initial resolution.

The first level of classification consisted of 428,963 points, of which 78,655 (18.3%) were manually annotated to be used as the training set and 77,663 (18.1%) as the evaluation set. The training of the model produced good results with a weighted average F1 score up to 0.96 (Table 2). Applying the classification to the whole dataset, approximately 22% of points under the support category were misclassified as statues. In the same way, some parts of the exposed pillars around the corner of the wall were misclassified (Figure 6 left); this is due to the fact that statues share a similar anisotropy feature (at a radius of 30 mm) with these elements.

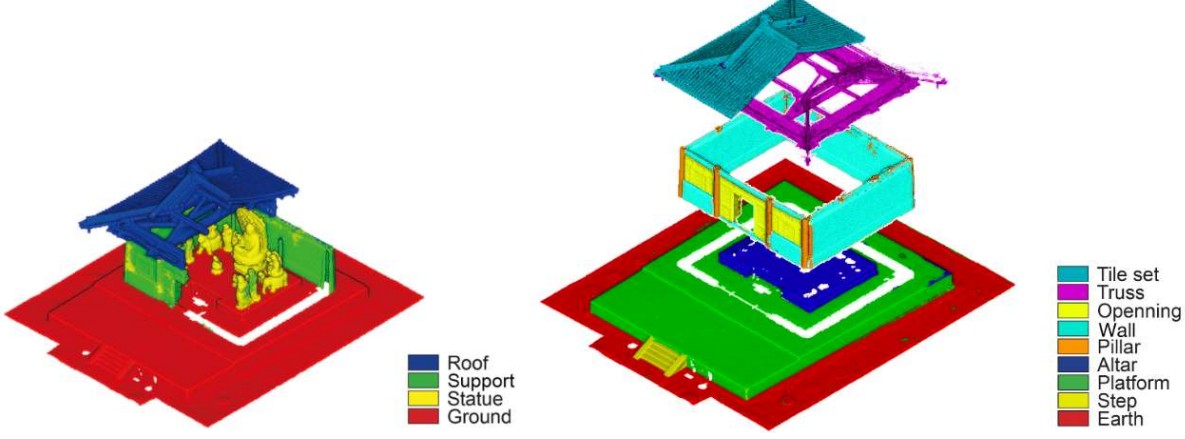

**Figure 6.** Classification results at level 1 (**left**) and level 2 (**right**) for NCS.

**Table 2.** Classification metrics at level 1 for NCS.

|  | Roof | Wall | Statue | Ground | WGT. Average |
|---|---|---|---|---|---|
| PREC. | 1.0 | 0.97 | 0.77 | 0.99 | 0.97 |
| RECALL | 1.0 | 0.78 | 0.97 | 0.99 | 0.96 |
| F1 | 1.0 | 0.86 | 0.86 | 0.99 | 0.96 |

The second level of classification on the roof, wall, and ground achieved satisfying results (Figure 6 right), with F1 scores up to 0.99, 0.93, and 0.99, respectively.

Classification level 3 makes use of normal vectors computed in the X and Y directions (at a search radius of 60 mm) as features to distinguish architectural elements, following their orientation. The scalar field of normal vectors in the x direction shows that they are associated with the corresponding labels (Figure 7). Scores from the evaluation set achieved a weighted average F1 score of 0.87 and generalized well to the whole dataset.

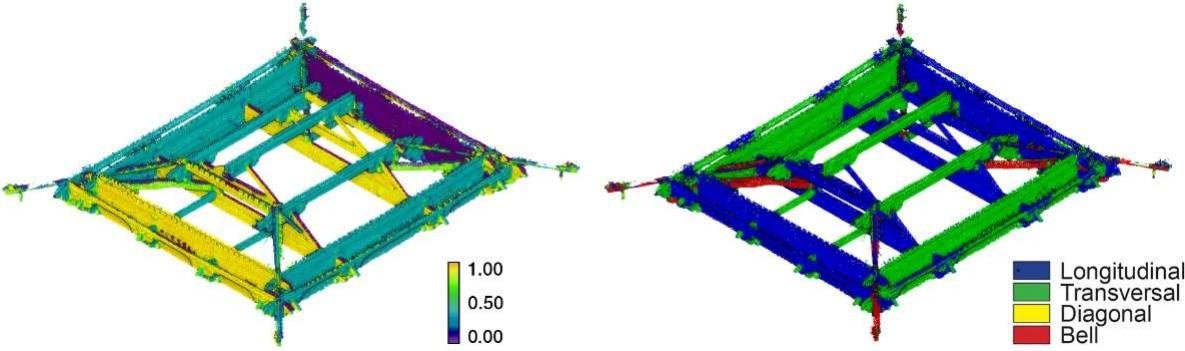

**Figure 7.** Normal vectors in the x direction (**left**) and labels (**right**) for the truss of the great hall of NCS.

### 4.2. The East Great Hall of Fokuang Ssu

The classification of the great hall is separated into three different levels (Figure 8). The first level refers to the basic framework of the hall, dividing the whole dataset into the roof, support, ceremonial object, and ground. The classification was carried out at a resolution of 100 mm in order to identify these categories.

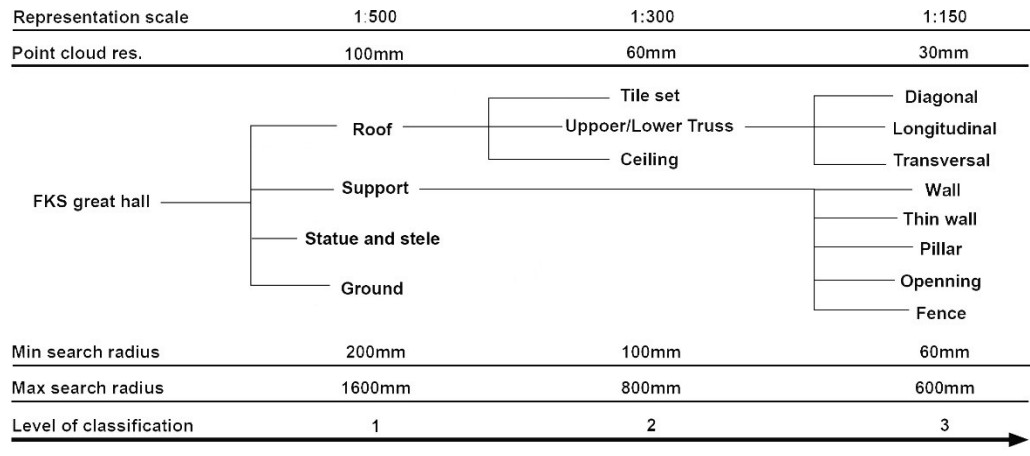

**Figure 8.** Point cloud classes for the FKS project.

The second level deepens these categories. The roof part is divided into tile set, truss, and ceiling. The support category comprises wall, thin wall, pillar, opening, and fence. The ceremonial objects include statues, steles, and some ritual objects. At this level, all datasets are processed at a 60 mm resolution.

This project uses the same geometric features calculated for the NCS dataset. These covariance features show, once more, satisfying correlations with the labels.

Classification level one consists of 624,049 points, with 51,028 (8.1%) points being manually annotated as the training set. The training model, tested on an evaluation set of 57,868 (9.2%) points, achieved satisfactory results (Figure 9).

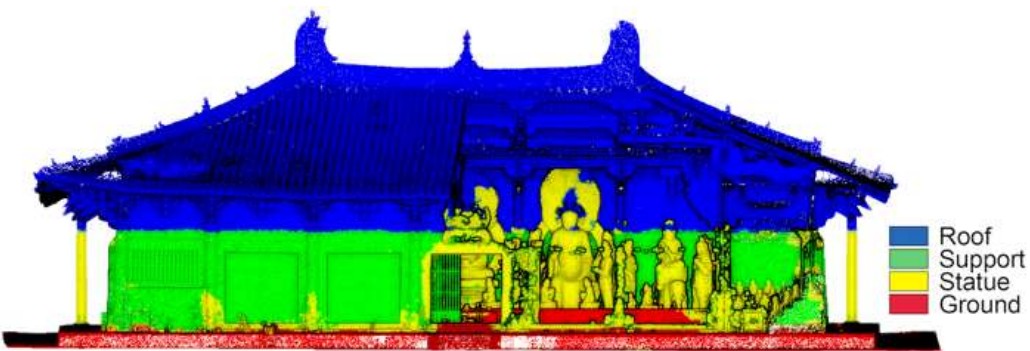

**Figure 9.** Classification result at level 1 for the east great hall in FKS.

The ground truth for the roof category contains 400,946 points, representing two-thirds of the dataset; this is why the classification model gives height-related features such as the Z coordinate high importance, in the order of 0.4235. The result is improved by adjusting the training sets (adding representative points from the nimbus part of the statue to the training set, not relying on Z coordinates to indicate the label), resulting in the importance of the Z coordinate being reduced to 0.3992 and F1 scores of up to 0.97 (Table 3).

**Table 3.** Classification metrics at level 1 for FKS.

|  | Roof | Support | Ceremonial | Ground | WGT. Average |
|---|---|---|---|---|---|
| PREC. | 0.99 | 0.96 | 0.85 | 0.98 | 0.97 |
| RECALL | 1.0 | 0.84 | 0.91 | 0.96 | 0.97 |
| F1 | 0.99 | 0.90 | 0.88 | 0.97 | 0.97 |

Classification level 2 (Figure 10 left) on the roof achieved an F1 score of 0.92 (Table 4). The model accurately illustrates how the roof structure is composed: below the tile sets, the wooden ceiling is mounted between the trusses.

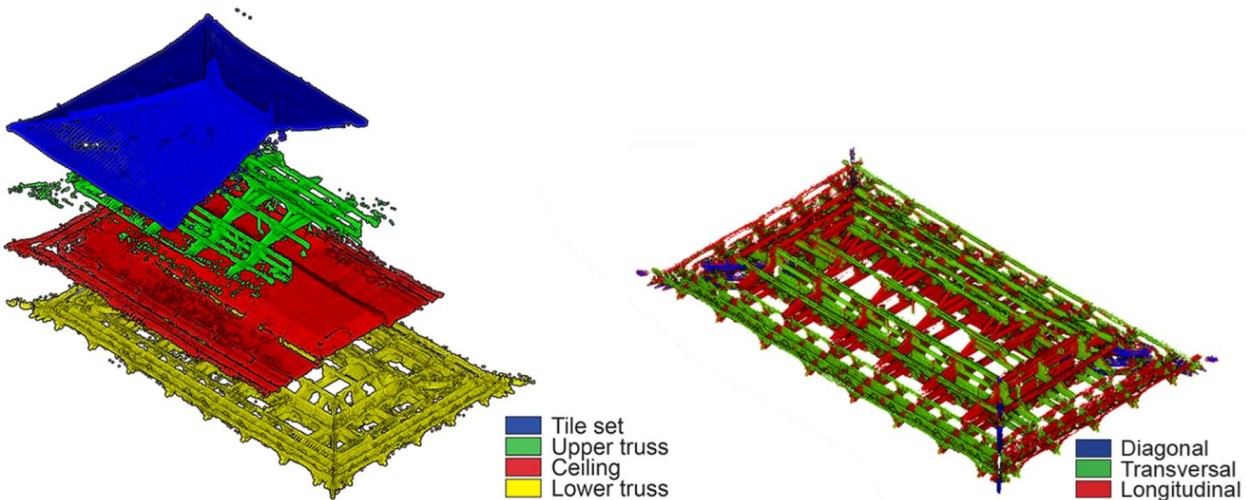

**Figure 10.** Classification result at level 2 on the roof (**left**) and level 3 on the truss (**right**) for FKS.

**Table 4.** Classification metrics at level 2 for FKS.

|  | Tile Set | Upper Truss | Ceiling | Lower Truss | WGT. Average |
|---|---|---|---|---|---|
| PREC. | 0.94 | 0.92 | 0.92 | 0.87 | 0.92 |
| RECALL | 0.92 | 0.82 | 0.96 | 0.93 | 0.92 |
| F1 | 0.93 | 0.87 | 0.94 | 0.90 | 0.92 |

Classification level three is aimed at distinguishing transversal from longitudinal and diagonal trusses. These architectonic components can be differentiated by their orientation. Using only geometric covariance features, it is impossible to tell which element is which inside the point model, as they all have the same geometric characteristics. However, normal vectors can distinguish the elements based on their orientation. Appending normal vectors in the X and Y directions, the classification model gains an accuracy of 0.89 when distinguishing diagonal, transversal, and longitudinal elements, and the result is visually clear (Figure 10). Providing a better scanned point cloud dataset of higher quality, classification could be performed to further recognize constructive components.

The classification performed on the supports at 30 mm resolution achieved an F1 score of 0.86. Pillars (including those partly laid in the wall), walls, slabs, ceilings, windows, and doors were distinguished. Some horizontal reinforcement rods behind the door slabs were misclassified, due to sharing the approximate surface of the wall.

### 4.3. The Bell Tower of Kaiyuan Ssu

Considering the monitoring needs and the original point cloud resolution and quality, the classification of the bell tower of Kaiyuan Ssu is subdivided into three different levels (Figure 11).

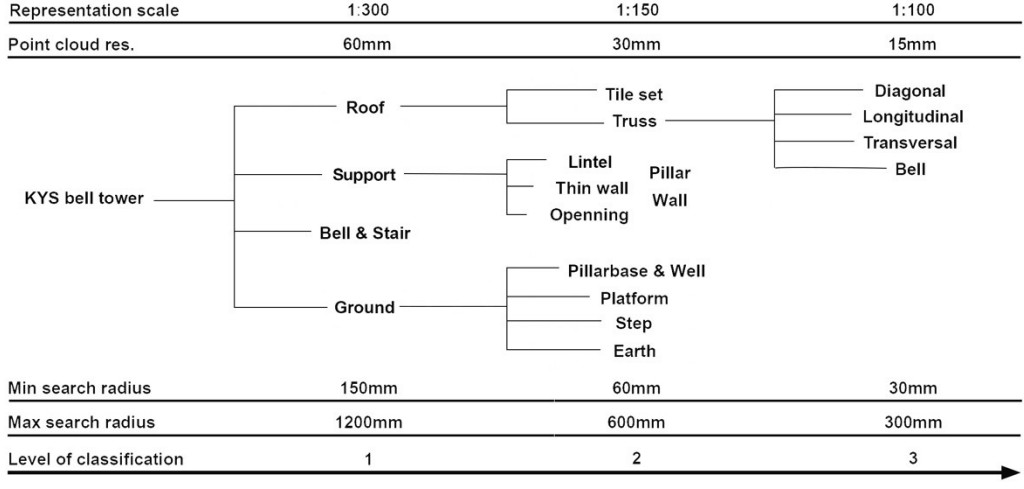

**Figure 11.** Point cloud classes for the KYS project.

The first level divides the whole dataset into roof, support, bell and stairs, and ground. To recognize these categories, the classification is performed on a 60 mm resolution subsampled version of the original dataset. The second level deepens these categories. The roof part is divided into the tile set and truss. The support category comprises the lintel, pillar, wall, brick wall, and opening subsets. The ground is divided into pillar bases and well, platform, steps, and earth.

The process at this intermediate step is performed on a 30 mm resolution copy of the full-resolution dataset. The third level of classification is performed to classify the truss into diagonal, longitudinal, and transversal elements and bells. The geometric features used to classify the dataset at this level comprise normal vectors in the X and Y directions, anisotropy, planarity, linearity, surface variation, sphericity, and verticality.

The first level of classification consists of 415,007 points, of which 100,366 (24%) and 106,085 (25%) were manually annotated to be used as the training set and evaluation set.

The training of the model produced good results (Figure 12, left), with a weighted average F1 score up to 0.91. The recall on the bell and stairs is 0.48. The second level of classification on the roof, support, and ground achieved satisfying results (Figure 12, middle), with F1 scores up to 0.95, 0.96, and 0.99, respectively. The last level of classification (level 3) distinguishes components belonging to diagonal, longitudinal, and transversal elements and small bells (Figure 12, right). The model achieved an overall accuracy of 0.90.

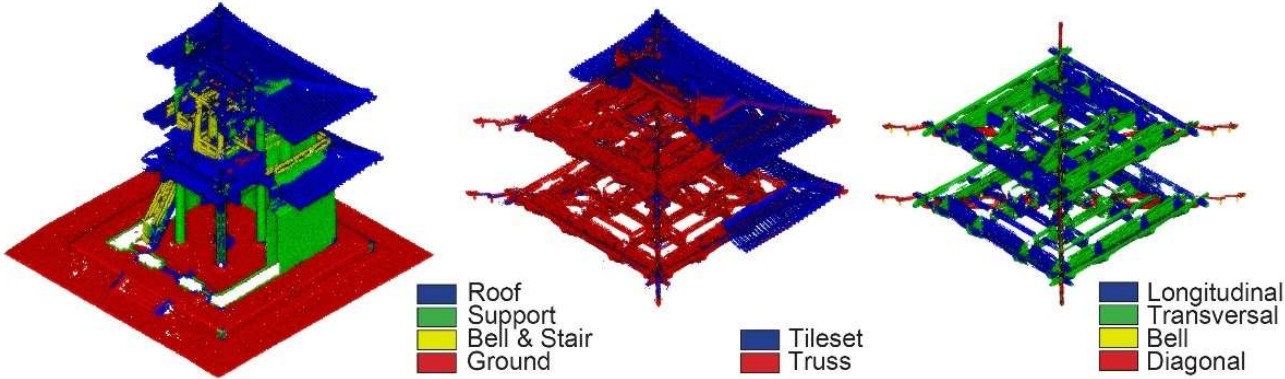

**Figure 12.** MLMR classification result at level 1 (**left**), level 2 on the roof (**middle**), and level 3 on the truss (**right**) for KYS.

## 5. Discussion

In most cases, numeric values are essential in order to mathematically evaluate the performance of a classification model, and visual representations of the results are irreplaceably illustrative.

Accuracy provides the direct score of the model and is commonly used to describe its performance. In addition, the confusion matrix, precision, recall, F1 score, and intersection over union can be calculated. These metrics are very important in highlighting the deficiencies of the trained model, especially in the case of overfitting. Weights of classes are commonly unbalanced, and therefore, visual presentation of the labels can be intuitive. It indicates the positions and portion of mis-classified points, where elements share similar geometric feature values of the mistaken labels. Through observing the scalar field of the results, the deficiency of the representative points or problems of features can be seen, especially when the weighted average F1 score is high while the recall score is low, or when the overall behaviour does not match the evaluation set score.

### 5.1. Approaches

The non-hierarchical classification approach is the most direct and is commonly used. It trains the model once with a training set that contains all of the labels necessary to describe the entire dataset. Since the features should be consistent with the categories, the training and prediction are performed on complicated and large datasets with feature computation ranging from small to large search radii. Under the same depth of trees, the performance is largely limited. The demand for computational power is high, and the final prediction presents great errors and misclassifications.

When applied to the great hall of Nanchan Ssu, a precision of 0.9332 is obtained (Figure 13). It took 21 s to train the model with 273,663 points, and 7 s to predict the dataset of 1,472,351 points. The results are encouraging, with an average F1 score of 93%. However, it cannot overcome the low score on the pillars (Table 5), as seen in the level 2 MLMR classification. When it is applied to predicting detailed and large datasets such as the east great hall of Fokuang Ssu, the performance is not optimal (Figure 13). Many points are not correctly predicted (Table 6). The prediction on the bell tower of Kaiyuan Ssu, having 20 classes, reaches 78% precision; however, a quarter of all categories cannot reach 0.50 recall.

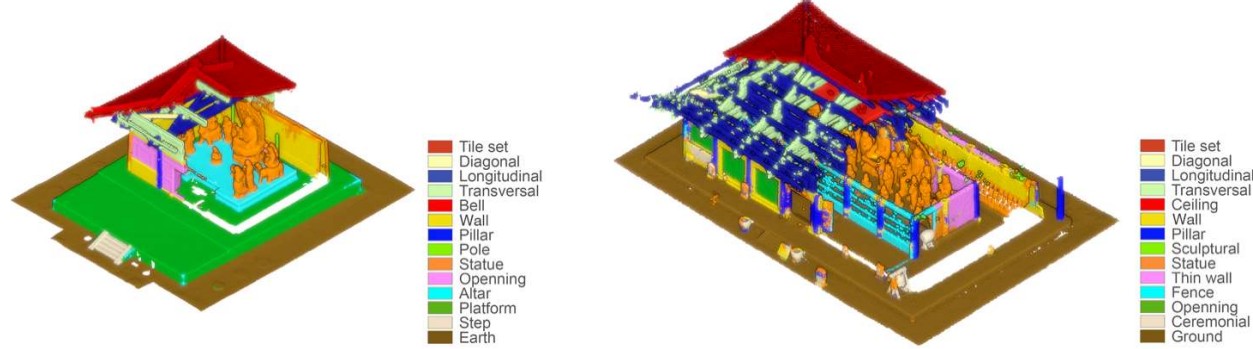

**Figure 13.** Non-hierarchical classification results for the NCS (**left**) and FKS (**right**).

**Table 5.** Non-hierarchical classification metrics for NCS.

| | Tile Set | Diag. Truss | Long. Truss | Transv. Truss | Bell | Wall | Pillar | Pole | Statue | Open. | Altar | Platf. | Step | Earth | WGT AVG |
|------|------|------|------|------|------|------|------|------|------|------|------|------|------|------|------|
| PREC. | 0.97 | 0.88 | 0.94 | 0.80 | 1.00 | 0.97 | 0.84 | 0.99 | 0.85 | 0.94 | 0.99 | 0.97 | 0.99 | 1.00 | 0.94 |
| REC. | 1.00 | 0.59 | 0.74 | 0.93 | 0.82 | 0.82 | 0.41 | 0.39 | 0.98 | 0.98 | 0.97 | 1.00 | 0.72 | 1.00 | 0.93 |
| F1 | 0.99 | 0.71 | 0.83 | 0.86 | 0.90 | 0.89 | 0.55 | 0.56 | 0.91 | 0.96 | 0.98 | 0.98 | 0.83 | 1.00 | 0.93 |

**Table 6.** RF non-hierarchical classification metrics for FKS.

| | Fence | Diag. Truss | Long. Truss | Transv. Truss | Ceiling | Tile Set | Wall | Sculpt. | Open. | Thin Wall | Cerem. | Pillar | Ground | Statue | WGT AVG |
|------|------|------|------|------|------|------|------|------|------|------|------|------|------|------|------|
| PREC. | 0.93 | 0.48 | 0.62 | 0.29 | 0.87 | 0.92 | 0.87 | 0.93 | 0.79 | 0.73 | 0.52 | 0.76 | 0.98 | 0.77 | 0.79 |
| REC. | 0.90 | 0.04 | 0.45 | 0.52 | 0.90 | 0.92 | 0.88 | 0.52 | 0.89 | 0.52 | 0.31 | 0.53 | 0.98 | 0.88 | 0.77 |
| F1 | 0.92 | 0.08 | 0.52 | 0.38 | 0.89 | 0.92 | 0.88 | 0.67 | 0.84 | 0.61 | 0.39 | 0.63 | 0.98 | 0.83 | 0.77 |

The MLMR approach performs classification at different levels and at different resolutions. The categories are distributed to each level following a predetermined tree-like structure. At each specific level, a particular model is trained using the training and evaluation sets as well as geometric features that are created and computed on the point model of that particular level. This avoids the heavy and redundant demand for representative points and features computation in the training process. After inheriting results from the prior classifications, each consequent work predicts merely part of previous datasets. Eventually, points gain labels that indicate layers of semantic meaning.

The cost of the two approaches (MLMR and non-hierarchical) differs based on the demand for computational resources. The post-processing for the point cloud in most cases demands a high-resolution dataset to achieve full detail. The computational cost of extracting geometric features for a full-resolution dataset is relatively high. Furthermore, these appended features will make the dataset even larger, resulting in a heavy burden for the training and prediction steps. In contrast to the non-hierarchical approach, in the MLMR approach, the computation of features, training, and prediction is conducted only on part of the original full-resolution dataset (where needed) or its subsampled copy. This avoids the generation of redundant geometric information and predictions. To be more specific, the computation of particular geometric features that highlight the edges and points that define tile sets is not necessary in distinguishing the roof part from other main parts of the building (e.g., walls, floors, etc.). The macro-category "roof" can be predicted at a relatively low resolution, where the roof objects appear geometrically homogeneous.

The time cost varies depending on the complexity of the projects. A lower number of categories or a less complex case study (which leads to less hierarchical levels nested) result in a similar duration for the training of the model and the prediction on the whole dataset for both multilevel multiresolution and non-hierarchical approaches. The manual intervention required from the user is in both cases quick and simple. Dealing with a simple, small-scale project, a non-hierarchical approach proves to be more effective. On the contrary, when

applied to a complex dataset with numerous layers of categories, the duration required for computing geometric features, training the model, and making predictions on the rest of the dataset increases exponentially. Therefore, the MLMR approach is more profitable, saving operational and computational costs in the training process, including preparing representative points, adjusting training features, and other manual adjustments.

*5.2. Generalization*

The MLMR approach brings with it another advantage. By splitting the classification labels over different layers with increasing detail, similar architectures with similar characteristics will have homogeneous representations of macro architectural components at the first level of classification, where only the main geometrical traits are important for the individuation of such elements (i.e., detailed elements are disregarded at this step and their classification is delegated to subsequent classification levels with higher resolution point models). A generalizability in the classification task is therefore not only possible but encouraged by the approach, leading to time and cost efficiency in the management of CH datasets. On the contrary, a single-step, non-hierarchical classification approach must recognize all elements at once, including enough samples for each representative element. It is impossible to find two different CH buildings with two identical elements when high accuracy must be maintained.

In order to test the generalizability of the approach, the point cloud models from NCS and FKS are used to create the training and evaluation sets on which the RF model is trained. Afterwards, the resulting classifier is used to spread the classification on the bell tower dataset (Kaiyuan Ssu).

A certain degree of pre-processing of the point cloud datasets is mandatory for focusing on the generalization tasks. The first step consists of defining the labels that are required to classify the unseen dataset of the belltower. Generalising the classification requires that the labels selected for the level 1 classification in the FKS and NCS datasets be the same of those in the KYS. Afterwards, all points not related to the construction itself (noisy points, moving peoples, and points not belonging to any defined label) are cleaned manually. In addition, NCS, FKS, and KYS datasets have all been subsampled at the same resolution (60 mm; the smallest resolution that allows to represent the smallest elements among level 1 classification objects, lowering the computational demands), and finally, they are translated in a similar local coordinate system (the x, y, and z coordinates have the same magnitude). Normal vectors in the x and y directions and covariance geometric features are computed on each dataset with the same search range radii.

5.2.1. Incompatibility of Classes and Representative Points

Usually, different datasets are represented by different groups of labels (classes of elements). At worst, the dataset to be classified features a different number and different types of classes than those included in the training set. If this is the case, trying to make a prediction on the unseen dataset, with a model trained with a different number of classes, will result in misclassified points. The model must assign a specific label to each point, and every label included in the model must be used. If that class is not present in the test set, this will result in a classification error (e.g., portions of walls classified as columns). As a result, recall and accuracy will undoubtedly be lowered. The same is true if the training set features fewer classes than the test set. It is therefore mandatory to define a standard set of labels that must be kept consistent among all datasets for which the classification (generalized) is performed.

Following the MLMR approach, the level 1 classification on the whole tower is first performed by the model trained with the training set (78,655 points, representing 18% of the dataset at 60 mm resolution) from the level 1 classification of the great hall of NCS. The categories of the training set comprise the roof, support, statue, and ground. The model obtained an accuracy of 0.95 on the evaluation set from NCS; however, it had an F1 score of only 0.62 on the bell tower of KYS (Figure 14 left). The problem is attenuated by not

using the Z coordinates (Figure 14 middle), a geometric feature which could be of high importance (0.4255). The obtained F1 score is 0.75. Visually, the results show that much of the error is due to the lack of representative points belonging to the pillar class in the training set built on NCS. For example, using the training set tailored on the FKS dataset, which contains individual instances of pillars, improves the performance of the model, gaining an accuracy of 0.75 (Figure 14 right); however, the model still cannot attribute pillars to the correct class (Tables 7 and 8).

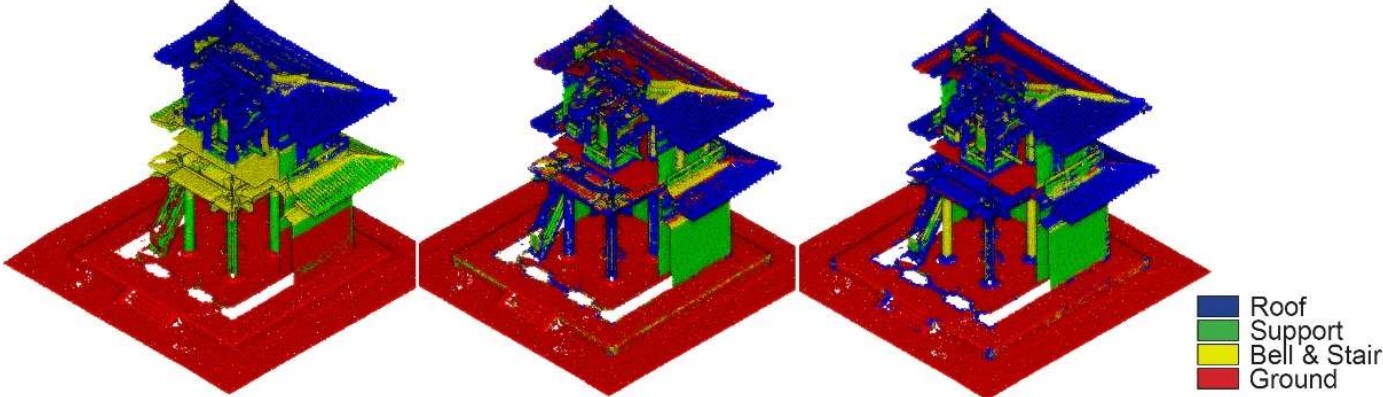

**Figure 14.** MLMR level 1 generalization test results for the KYS using training sets from NCS with the Z coordinate (**left**) and without (**middle**), and the training set from FKS (**right**).

**Table 7.** Classification metrics for the generalization test on KYS at level 1 using the training set from NCS without a Z coordinate.

|  | Roof | Support | Bell and Stair | Ground | WGT. Average |
|---|---|---|---|---|---|
| PREC. | 0.83 | 0.81 | 0.14 | 0.74 | 0.77 |
| RECALL | 0.81 | 0.57 | 0.22 | 0.94 | 0.74 |
| F1 | 0.82 | 0.67 | 0.17 | 0.83 | 0.75 |

**Table 8.** Classification metrics for the generalization test on KYS at level 1 using the training set from FKS without a Z coordinate.

|  | Roof | Support | Bell and Stair | Ground | WGT. Average |
|---|---|---|---|---|---|
| PREC. | 0.81 | 0.77 | 0.24 | 0.80 | 0.77 |
| RECALL | 0.85 | 0.65 | 0.20 | 0.92 | 0.65 |
| F1 | 0.83 | 0.70 | 0.22 | 0.85 | 0.77 |

The level 2 generalization test on the roof part is performed with a training set generated from the Nanchan Ssu (NCS) roof part, which comprises 147,708 points (taking up 15% of the roof dataset at 30 mm resolution). The model gains an accuracy of 0.95 on the evolution set of the same building and reaches promising results (Figure 15 left) on the roof part of KYS with an accuracy of 0.85 (Table 9).

**Table 9.** Classification metrics for the generalization test on the roof part of KYS at level 2 using the training set from NCS without a Z coordinate.

|  | Tile Set | Truss | WGT. Average |
|---|---|---|---|
| PREC. | 0.80 | 0.94 | 0.86 |
| RECALL | 0.96 | 0.72 | 0.85 |
| F1 | 0.87 | 0.82 | 0.85 |

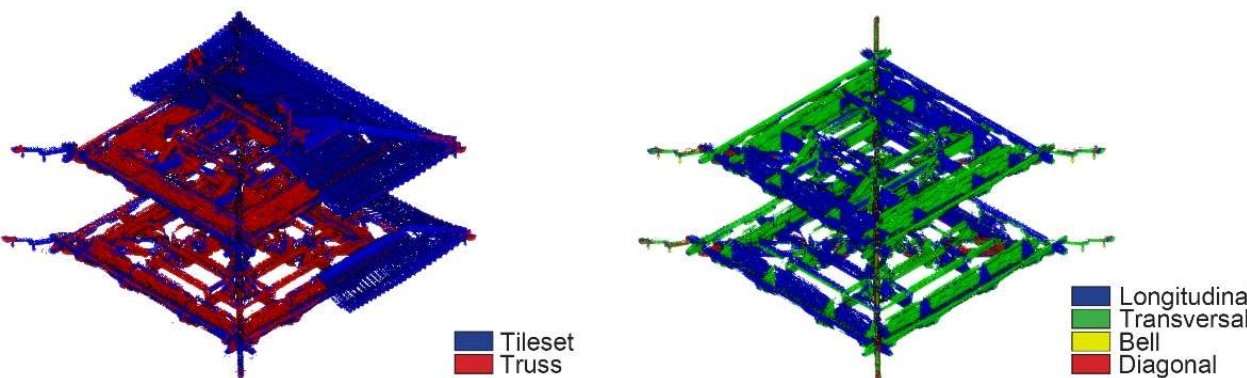

**Figure 15.** MLMR level 2 (**left**) and 3 (**right**) generalization test results for the KYS using training sets from NCS.

The level 3 generalization test on the truss is performed by the model trained with the training set, selected from the truss part of NCS (130,344 points, 10% of the truss dataset at 15mm resolution). The result (Figure 15 right) on the evaluation set of the same building is 0.88, while on the test dataset (Kaiyuan Ssu, KYS), it is 0.74 (Table 10).

**Table 10.** Classification metrics for the generalization test on the truss part of KYS at level 3 using the training set from FKS without a Z coordinate.

|  | Long. | Transv. | Bell | Diagonal | WGT. Average |
|---|---|---|---|---|---|
| PREC. | 0.70 | 0.79 | 0.98 | 0.73 | 0.75 |
| RECALL | 0.83 | 0.76 | 0.52 | 0.43 | 0.74 |
| F1 | 0.76 | 0.77 | 0.68 | 0.54 | 0.74 |

5.2.2. Regarding Resolution

Different datasets also differ by their resolution. This affects the results mainly in the process of computing geometric features. Datasets at a higher resolution can perform computation with a wide range of search radii, starting from a tolerable least mean resolution (an average local neighbourhood radius). Computation at a lower resolution and with a relatively smaller search radius will generate many NaN (not a number) values, which will be redundant and confusing for the random forest algorithm. If the trained model is based on a training set with a higher resolution, the prediction cannot rely on these features with NaN values. When the training set is at a much lower resolution than the dataset to be predicted, the behaviour will not be optimal.

Generalization tests were conducted using the level 1 training set and evaluation set from the NCS MLMR classification, at a 60 mm resolution (Figure 16, Table 11). Six test sets were subsampled from the KYS dataset at resolutions of 30, 40, 60, 90, 120, and 150 mm (resolutions upon which labels are still definable, while halving or doubling the number of points starting from the initial resolution). Geometric features were computed separately for each test set. The model appeared to have slightly better results when the resolution was closer to that of the training set. In the same way, six test sets from FKS were tested (Table 12), and the results showed the same trend, with performance increasing as the resolution of the test set becomes closer to that of the training set. While comparing the time costs of this generalization test, the computation of geometric features on a 30 mm resolution test set will take six times the duration of a 60 mm resolution set, and four times for prediction. Considering minor differences in the overall behaviour, the higher-resolution datasets are subsampled so as to achieve efficient processing time and computational resources (Figure 17).

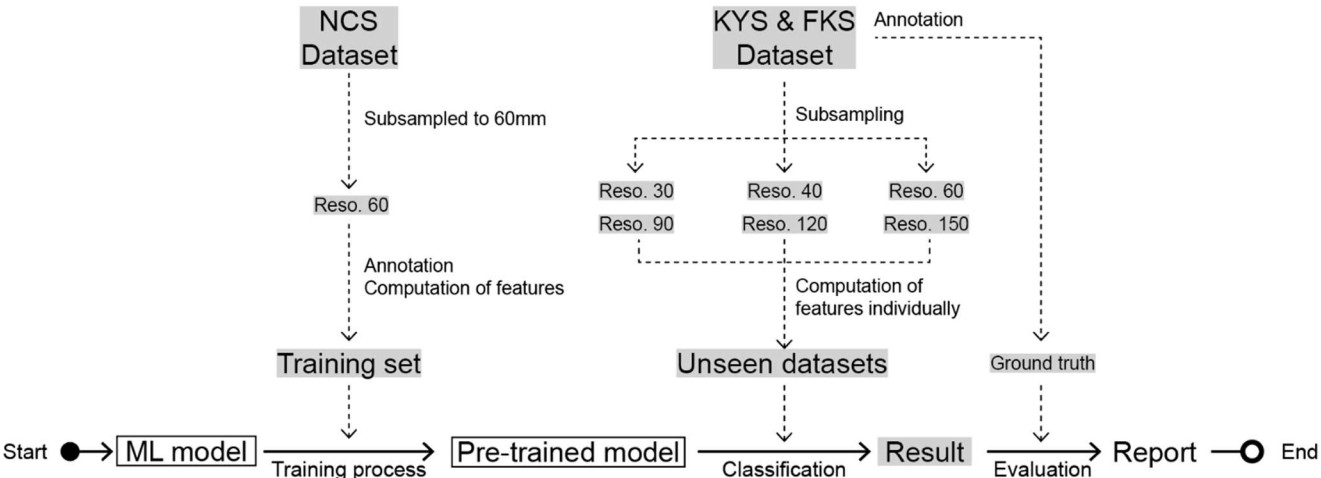

**Figure 16.** Classification regarding the resolution in the generalization.

**Table 11.** Generalization tests of resolution on KYS.

| Reso. | Points | Acc. on Eval. Set | Acc. on KYS | Mac. Avg. F1 | WGT. Avg. F1 |
|-------|-----------|-----------------|-------------|--------------|--------------|
| 30 | 1,603,902 | 0.86801 | 0.72 | 0.61 | 0.73 |
| 40 | 705,033 | 0.86801 | 0.73 | 0.61 | 0.74 |
| 60 | 347,703 | 0.86801 | 0.74 | 0.62 | 0.75 |
| 90 | 184,493 | 0.86801 | 0.73 | 0.61 | 0.74 |
| 120 | 103,123 | 0.86801 | 0.72 | 0.60 | 0.74 |
| 150 | 60,368 | 0.86801 | 0.70 | 0.59 | 0.73 |

**Table 12.** Generalization tests of resolution on FKS.

| Reso. | Points | Acc. on Eval. Set | Acc. on FKS | Mac. Avg. F1 | WGT. Avg. F1 |
|-------|-----------|-----------------|-------------|--------------|--------------|
| 30 | 5,015,970 | 0.95159 | 0.88 | 0.74 | 0.88 |
| 40 | 3,193,979 | 0.95159 | 0.88 | 0.74 | 0.88 |
| 60 | 1,590,049 | 0.95159 | 0.88 | 0.74 | 0.88 |
| 90 | 754,806 | 0.95159 | 0.88 | 0.74 | 0.89 |
| 120 | 433,866 | 0.95159 | 0.88 | 0.74 | 0.89 |
| 150 | 278,527 | 0.95159 | 0.88 | 0.73 | 0.88 |

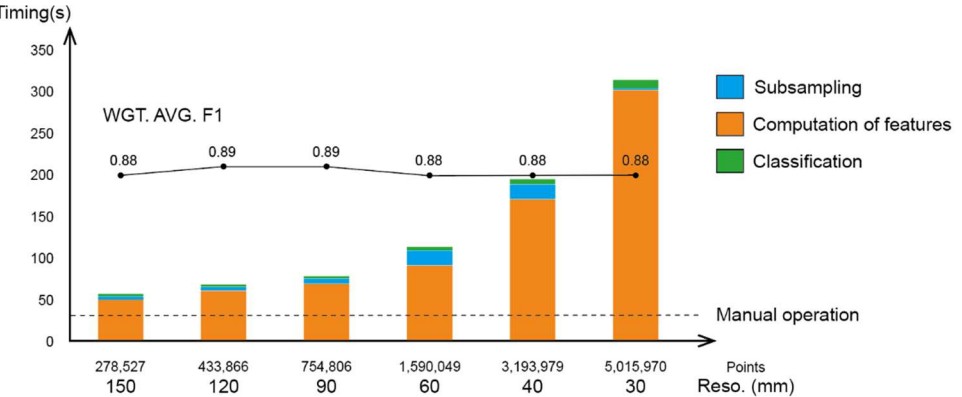

**Figure 17.** Timing of the generalization tests for resolution on FKS.

### 5.2.3. General Pre-Trained Model

A general pre-trained model might be able to maximize the applicability of the MLMR approach. Being trained with representative points selected from various datasets, the pre-trained model is expected to recognize corresponding elements. A combined training

set could outperform a single-source set, due to the fact that the combined supplement can help to better define the categories, compensating for any deficiency.

In the MLMR level 1 generalization test, appending the representative points of pillars from FKS to the training set of NCS can improve the model behaviour on the bell tower of KYS (Figure 18). This means generating a completely new training set, which takes representative elements for each label from different datasets to classify an unseen point model. The accuracy increased to 0.77 (Tables 13 and 14), and most importantly, the points of the pillars were classified under the correct categories (Figure 19).

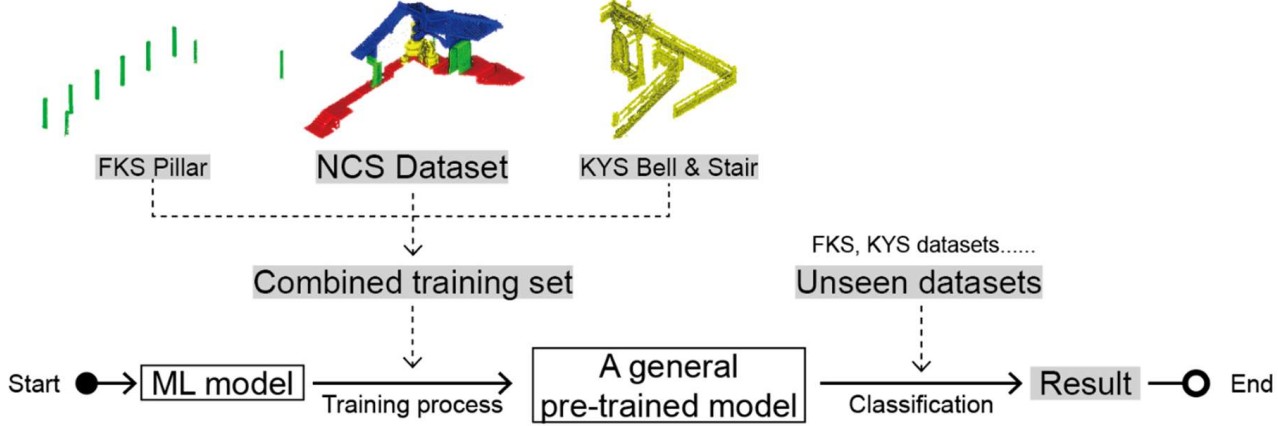

**Figure 18.** Pipeline for the general pre-trained model test.

**Table 13.** Classification metrics for the generalization test on KYS at level 1 using a training set of NCS and combining points of pillars from FKS.

|  | Roof | Support | Bell and Stair | Ground | WGT. Average |
|---|---|---|---|---|---|
| PREC. | 0.86 | 0.79 | 0.14 | 0.75 | 0.78 |
| RECALL | 0.80 | 0.71 | 0.18 | 0.94 | 0.77 |
| F1 | 0.83 | 0.75 | 0.16 | 0.83 | 0.77 |

**Table 14.** Classification metrics for the generalization test on FKS at level 1 using a training set of NCS and combining points of pillars from FKS.

|  | Roof | Support | Statue | Ground | WGT. Average |
|---|---|---|---|---|---|
| PREC. | 0.99 | 0.82 | 0.39 | 0.82 | 0.90 |
| RECALL | 0.97 | 0.80 | 0.38 | 0.97 | 0.90 |
| F1 | 0.98 | 0.81 | 0.38 | 0.89 | 0.90 |

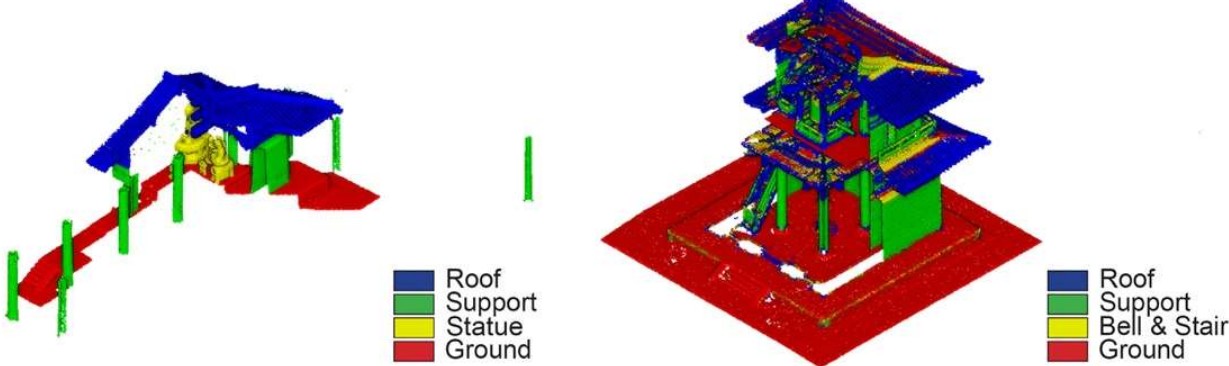

**Figure 19.** Training set (**left**) where points from NCS and FKS coincide and result (**right**) for the generalization test.

The test indicated the applicability of a general pre-trained model by combining the annotated portion of the point clouds with the corresponding geometric features. The test demonstrated that, if provided with more specific representative points of similar elements of the bell and stairs, the results on KYS will be improved (Table 15); however, the model will have a lower score on the original datasets, as the training set is not specifically tailored to fit them (Table 16).

**Table 15.** Classification metrics for the generalization test on KYS at level 1 using a training set of NCS and combining points of the pillars from FKS and the bell and stairs from KYS.

|  | Roof | Support | Bell and Stairs | Ground | WGT. Average |
|---|---|---|---|---|---|
| PREC. | 0.90 | 0.91 | 0.24 | 0.78 | 0.84 |
| RECALL | 0.75 | 0.59 | 0.85 | 0.94 | 0.75 |
| F1 | 0.82 | 0.72 | 0.38 | 0.85 | 0.77 |

**Table 16.** Classification metrics for the generalization test on FKS at level 1 using a training set of NCS and combining points of the pillars from FKS and the bell and stairs from KYS.

|  | Roof | Support | Statue | Ground | WGT. Average |
|---|---|---|---|---|---|
| PREC. | 1.00 | 0.87 | 0.28 | 0.85 | 0.91 |
| RECALL | 0.90 | 0.75 | 0.57 | 0.96 | 0.86 |
| F1 | 0.94 | 0.80 | 0.38 | 0.90 | 0.88 |

*5.3. Features*

Features address the heterogeneity in point clouds; after training, they gain correlations with labels. High-quality features allow the better interpretation of the models and the enhancement of the algorithm performance concerning both speed and accuracy.

The coordinates in the training process are the most basic for point clouds. The X and Y coordinates suggest the horizontal planar projections of points, and the Z coordinate suggests the height. In most training processes, the three coordinates gain high importance, which results in overfitting. However, in cases where elements are non-repetitive in the vertical direction, such as single-floor architectures (e.g., the NCS great hall) and projects for which the elements vary with verticality, the Z coordinate is the most profitable feature. However, when the same model is generalized on multistorey objects (such as the KYS bell tower), this feature will result in overfitting (as mentioned in the section Generalization).

Geometric covariance features represent the inter-point correlation in a certain radius, and their computation is indispensable for the classification approach described. Apart from certain types of numerical correlation, an equally essential property is the search radius. This represents the local neighbourhood; computation with a too-large search radius will exclude detailed geometric information, while a small one will produce non-numeric results (points out of reach for each) depending on the dataset resolution. Different radii should be considered in the computation of geometric features, the combination of them helping the algorithm to make better decisions and to better generalize the results.

Normal direction vectors can help when distinguishing the objects that are highly dependent on their orientation. When it comes to the classification of roof trusses in Chinese wooden architectures, even if other geometric features are confusing, normal directions contribute to the improvement of the results. When performing prediction on points from large ranges of planar areas (for instance, the MRML level 2 classification of the wall and ground of the KYS bell tower at 30 mm resolution), using normal vectors in X and Y direction in the training process results in overfitting.

**6. Conclusions and Future Works**

This study presents and discusses the behaviour of a hierarchical classification procedure, multilevel multiresolution (MLMR), on singular projects and its generalization across different Chinese CH complexes: the great hall of NanChan Ssu (NCS), the east great hall

of FoKuang Ssu (FKS), and the bell tower of Kaiyuan Ssu (KYS). The three buildings vary in terms of scale, complexity, and architecture type.

The experiments demonstrate that a classification performed following a non-hierarchical approach is effective when the project is relatively small and simple and has a low number of class subdivisions. The one-step classification reaches similar results to those obtained with the MLMR approach, but it requires less manual labour and less elaboration time.

The hierarchical approach (MLMR) shows its true potential on the types of architecture that encompass different materials and scales among the elements of the buildings, and whose dimensions are monumental. The dataset is subdivided into a hierarchy of nested class labels with increasing detail, and a specific classification step is performed at each layer. The approach requires more manual adjustment in terms of the preparation of the dataset and training set for each classification task, but computational resources used for extracting geometric features are largely saved, as only a portion of the dataset is processed at full resolution. In addition, MLMR allows manual adjustment on each level, providing a hierarchical semantic label tree for point cloud management.

Generalization tests may indicate that the MLMR approach has a wide range of applicability, if the provided categories are instructively clear. The performance of a machine learning classification depends both on the training set (in terms of quality) and on how the model is trained. These two aspects are decisive and highly influence the results, with regard to labelled classes, accuracy, etc.

From the tests above, given clearly identified tasks, the model produces promising results. However, generally, the training set and the test set do not match in terms of categories. This is to say, two different datasets have different sets of semantic labels. When it comes to a great variety of architecture projects, it is demanded that the categories in each corresponding level are consistent and have a corresponding resolution. In the meantime, the training set should always include an adequate number of representative points for the classifier to be able to correctly label all architectural elements.

MLMR classification can achieve satisfying results in specific conditions. The approach generalizes well when testing and training sets are collected from the same type of architecture. It can discern common architectonic components following a semantic hierarchy. The manual adjustments are both allowed and necessary, for removing irrelevant points and reassigning misclassified points.

Expert intervention is still mandatory to design label categories and to prepare representative training sets to pre-train the model. The 3D documentation process should guarantee that the quality of the point cloud can meet the classification needs (completeness, resolution, and noise). In this work, the pre-trained model achieved a weighted average F1 score of least 0.74, and is expected to gain higher F1 scores, ranging from 80% to 90%, with the combined training sets. Considering the operational cost, this approach is effective for practical use in point cloud post-processing.

Future works should be based on more datasets, testing the categories, and generalization, with the creation of a generic framework of categories that may be used with the models to perform corresponding predictions. The MLMR and generalization processes will benefit from instructively clear categories, which can be used to train the model properly and to recognize elements without ambiguity. The naming of labels should be built upon further ontological and empirical studies of geometric features, reducing information discrepancies, inconsistencies, and errors [20].

Geometric features, normal vectors, and Z coordinates prove efficient in classification. However, the choice of which feature to use during the training of the model is very important. The model could easily overfit its training set, resulting in a totally incorrect prediction. Improving the quality of the point cloud in terms of physical attributes and mean resolution will allow more advanced classifications. Specifically, a fully covered point cloud (roof top, truss, and narrow spaces) with colour information would make the architectural elements more distinguishable.

In general, the MLMR approach is a useful machine learning tool dealing with classification works on single point cloud dataset. In this approach, it is demonstrated that the pre-trained model has satisfying generalization capability and can be used to process unknown test sets.

**Author Contributions:** Conceptualization, K.Z., S.T. and F.F.; methodology, K.Z. and S.T.; software, K.Z.; validation, K.Z. and S.T.; formal analysis, K.Z. and S.T.; investigation, K.Z. and Y.D.; resources, Y.D.; data curation, K.Z.; writing—original draft preparation, K.Z.; writing—review and editing, K.Z., S.T., Y.D. and F.F.; visualization, K.Z.; supervision, Y.D. and F.F.; project administration, F.F.; funding acquisition, K.Z. All authors have read and agreed to the published version of the manuscript.

**Funding:** This research was funded by the China Scholarships Council, grant number 202208520007.

**Institutional Review Board Statement:** Not applicable.

**Informed Consent Statement:** Not applicable.

**Data Availability Statement:** The data presented in this study are available upon request from the corresponding author. The data are not publicly available due to the Regulations for the Implementation of the Cultural Relics Protection Law of the People's Republic of China.

**Acknowledgments:** Financial support from the program of the China Scholarships Council (grant number: 202208520007) is acknowledged. We thank all contributors from the Archaeology Centre for Architecture, Settlement and Landscape (ACASL), Tianjin University, China, for the help during the investigation and collection of point cloud datasets.

**Conflicts of Interest:** The authors declare no conflict of interest.

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
