# Peer review of "A Multilevel Multiresolution Machine Learning Classification Approach: A Generalization Test on Chinese Heritage Architecture"

_heritage, doi:10.3390/heritage5040204_

Round 1

Reviewer 1 Report

Ref not found at lines 32, 87, 184, 190, 683.

Author Response

Reviewer 1

Ref not found at lines 32, 87, 184, 190, 683.

Reply to reviewer 1:

Dear reviewer:

Thank you for the review and comment.

We are sorry for the trouble with citations (in lines 32, 87, 184, 189, 193); it may be caused by the crossreference issue of Microsoft word.

We have corrected them and made some other complements. You may check them in the revised version of the paper.

Reviewer 2 Report

Dear authors,

I want to thank you for your efforts in producing this interesting paper.

The paper addresses a relevant topic of the machine learning classification approach tested on several ancient wooden structures. The article is well written; I have only a few comments below.

Below is a list of proposed amendments or comments on the manuscript content.

Line 32: missing citation; please check.

Line 87: missing citation; please check.

Line 184: again, missing citation; please check.

Line 189: again, missing citation; please check.

Line 193: missing citation.

Line 177-194: please specify the parameters set during the scanning or at least some quality parameters for the points of the point clouds, e.g., the accuracy in the spatial position of a single measurement point.

Lines 272-273: "the original point clouds have been subsampled" I understand that in some cases, the original point cloud has (unnecessary) high density, so only a theoretical question, did you try the whole classification procedure on the original (not downsampled) point clouds? Is this subsampling somehow affecting the results? Thank you for your answer.

Line 289: based on what were radii chosen? Some assumptions?

Line 296: "the original point cloud resolution and quality" the quality of the point clouds was not mentioned previously in the article; it would be appropriate to add this quality information when listing the point clouds for the case studies in chapter 3.

Figure 8: Please specify based on what were these min and max search radii chosen.

Lines 424-425: there is some strange break of the lines; please check.

Lines 488-489: "all points that are not related to the construction itself (noisy points, moving peoples, points not belonging to any defined label) are cleaned" how? Are they filtrated manually or somehow automatically?

Line 491: why resolution 60mm? Can the authors justify it?

Line 559: "resolution of 30, 40, 60, 90, 120, 150mm" how were these values chosen, based on the author's assumptions? Or is it dependent on the data tested?

Author Response

Dear authors,

I want to thank you for your efforts in producing this interesting paper.

The paper addresses a relevant topic of the machine learning classification approach tested on several ancient wooden structures. The article is well written; I have only a few comments below.

Below is a list of proposed amendments or comments on the manuscript content.

Line 32: missing citation; please check.

Line 87: missing citation; please check.

Line 184: again, missing citation; please check.

Line 189: again, missing citation; please check.

Line 193: missing citation.

Line 177-194: please specify the parameters set during the scanning or at least some quality parameters for the points of the point clouds, e.g., the accuracy in the spatial position of a single measurement point.

Lines 272-273: "the original point clouds have been subsampled" I understand that in some cases, the original point cloud has (unnecessary) high density, so only a theoretical question, did you try the whole classification procedure on the original (not downsampled) point clouds? Is this subsampling somehow affecting the results? Thank you for your answer.

Line 289: based on what were radii chosen? Some assumptions?

Line 296: "the original point cloud resolution and quality" the quality of the point clouds was not mentioned previously in the article; it would be appropriate to add this quality information when listing the point clouds for the case studies in chapter 3.

Figure 8: Please specify based on what were these min and max search radii chosen.

Lines 424-425: there is some strange break of the lines; please check.

Lines 488-489: "all points that are not related to the construction itself (noisy points, moving peoples, points not belonging to any defined label) are cleaned" how? Are they filtrated manually or somehow automatically?

Line 491: why resolution 60mm? Can the authors justify it?

Line 559: "resolution of 30, 40, 60, 90, 120, 150mm" how were these values chosen, based on the author's assumptions? Or is it dependent on the data tested?

Reply to the reviewer:

Dear reviewer:

We want to thank you for your comments and detailed suggestions.

We are sorry for the trouble with citations (in lines 32, 87, 184, 189, 193), it may be caused by the crossreference issue of Microsoft word.

For comments on lines 177-194: Some original files and settings cannot be retrieved, because the investigations were conducted several years ago. The datasets include scans from the whole temple, and settings vary from scene to scene. For instance, points per scan rise for the statues and lower for the surroundings. The operators did this in this way to save time on site, while the qualities of the point clouds were guaranteed to suit the monitoring needs (resolution 3-10mm).

Lines 272-273: The subsampling affects the results. Usually, the resolution of the original point cloud is not globally even, it will result in deviation in the computation process of geometric features and eventually confuse the classification. Also, subsampling is mandatory in those situations where the computational demands are too high due to the point numerosity. However in the MLMR approach, the classification starts from lower-resolution datasets,  and only part of the original point clouds is used for specific and detailed classification tasks, where full resolution is needed to represent fine details.

Line 289: The radii were chosen concerning the measurements of under-defining elements. The set of radii comprises a wide range of them to better describe the local point features.

Line 491: 60mm is the same resolution used in the MLMR level 1 classification on KYS, to make  comparisons, while it’s also the smallest resolution that allows representing the smallest elements among level 1 classification objects, lowering the computational demands

Line 559: The subsampling aims to generate datasets of various resolutions and test the model on them, starting from 60mm resolution one, we chose resolutions upon which the generated features can still define the elements while half or twice the number of points, as it's helpful to demonstrate the conclusion that resolution affects the performance slightly, but has a great impact on time consumption.

We have corrected the paper and made some complements. You may check them in the revised version of the paper.

Reviewer 3 Report

The paper is poorly written. There are missing references with clear error messages in the text. The English is very bad, the words are missing letters, the sentences are sometimes hard to understand.

There are some claims that introduce some doubt, e.g. how ResNet as deep neural network model proves the potential of cloud point dataset?

The authors refer to CH shortcut. It is never explained.

The aforementioned  linguistic defects lead to the basic problem of the paper - what is the aim of the work? To reduce the number of annotations needed to perform for each structure? This is not clearly stated anywhere in the text. After a short introduction (very short, briefly describing different methods), the authors provide description of the analyzed buildings and jump straight into the solution. Once again, what needs to be done? Why it needs to be done? How manual annotation lowers the quality o the samples (as stated in the introduction)?

Furthermore Figure 4 suggests, that KYS bell tower will be automatically annotated using model trained on part of NCS cloud points. In chapter 4.3 the authors state however that the model was trained with part of the KYS bell tower points that were automatically annotated. So once again what was the aim here? To create general model to annotate Tang dynasty styled buildings or to limit the manual annotation work needed to be done for each building?

The generalization is to some extend discussed in chapter 5.2. More testes would be however helpful in establishing how the general model trained using NCS dataset will fork not only on KYS but also on FKS and vice versa. That perhaps would show how the universal dataset should be combined to introduce all needed elements into the training set. Exploring this would greatly improve the value of the paper.

Author Response

The paper is poorly written. There are missing references with clear error messages in the text. The English is very bad, the words are missing letters, the sentences are sometimes hard to understand.

There are some claims that introduce some doubt, e.g. how ResNet as deep neural network model proves the potential of cloud point dataset?

The authors refer to CH shortcut. It is never explained.

The aforementioned  linguistic defects lead to the basic problem of the paper - what is the aim of the work? To reduce the number of annotations needed to perform for each structure? This is not clearly stated anywhere in the text. After a short introduction (very short, briefly describing different methods), the authors provide description of the analyzed buildings and jump straight into the solution. Once again, what needs to be done? Why it needs to be done? How manual annotation lowers the quality o the samples (as stated in the introduction)?

Furthermore Figure 4 suggests, that KYS bell tower will be automatically annotated using model trained on part of NCS cloud points. In chapter 4.3 the authors state however that the model was trained with part of the KYS bell tower points that were automatically annotated. So once again what was the aim here? To create general model to annotate Tang dynasty styled buildings or to limit the manual annotation work needed to be done for each building?

The generalization is to some extend discussed in chapter 5.2. More testes would be however helpful in establishing how the general model trained using NCS dataset will fork not only on KYS but also on FKS and vice versa. That perhaps would show how the universal dataset should be combined to introduce all needed elements into the training set. Exploring this would greatly improve the value of the paper.

Reply to the reviewer:

Dear reviewer,

We thank you for your detailed comments and suggestions.

First of all, apologize for the missed shortcuts and cross-reference issue of wrong citations.

For the ResNet in the content, it refers to previous research that has shown the potential of AI in natural language processing (1D) and image (2D) classification fields, the AI technologies are expected to play also an important role in 3D point cloud classification fields with different network architectures. In other words, ResNet proves the potential of AI on 1D and 2D datasets and only indirectly on point cloud datasets.

Segmentation and annotation are important for better utilization of the 3D point cloud assets. The common manual approach is criticized because it’s time-consuming and subjective (since labels have different annotation criteria and each should be consistent). We expected that AI technology can predict the label to save time and human cost. In the end, it did the tasks quickly and correctly. In this paper, we discussed if the trained AI model can generalize well also on other unseen datasets, for which we had positive answers.

For your concern about the content, chapter 3 introduced the results of the MLMR classification approach on 3 datasets, while in chapter 4 we had a brief discussion on them and on how the trained model generalizes on unseen datasets.

We agree with your opinion on conducting more generalization tests, as was mentioned in the final chapter. We hope to establish a general model and training strategy that can be used in practical scenes in the future, it takes extra effort but the results should be promising.

We have made some complements to the paper that you may check in its revised version.

Reviewer 4 Report

Article describing an interesting approach for the semantic segmentation of the point cloud as a further contribution to the research work that has been carried out. Difference is made here for the case study, e.g. the architectural typology compared to the case studies mentioned in the article. Clear and straightforward presentation of content. Discussions logically explained and correctly motivated.

I invite the authors, as I presume they do, to check the form of the text, there are some oversights: capital letters to be corrected e.g. title of paragraph 2.1 and line 648; in the caption of figure 14 the reference to the figure on the right is missing (insert "right") and revise the errors concerning the links of the references. 

Author Response

Article describing an interesting approach for the semantic segmentation of the point cloud as a further contribution to the research work that has been carried out. Difference is made here for the case study, e.g. the architectural typology compared to the case studies mentioned in the article. Clear and straightforward presentation of content. Discussions logically explained and correctly motivated.

I invite the authors, as I presume they do, to check the form of the text, there are some oversights: capital letters to be corrected e.g. title of paragraph 2.1 and line 648; in the caption of figure 14 the reference to the figure on the right is missing (insert "right") and revise the errors concerning the links of the references.

Reply to the reviewer:

Dear reviewer,

thank you for the review and detailed suggestions.

We are sorry for the miss citations, caption issue and other format problems. We have corrected them and made some other complements. You may check them in the revised version of the paper.

Round 2

Reviewer 3 Report

Thank you for the corrections, all my remarks were taken into account.